# Benefits of Chronic Administration of a Carbohydrate-Free Diet on Biochemical and Morphometric Parameters in a Rat Model of Diet-Induced Metabolic Syndrome

**DOI:** 10.3390/metabo13101085

**Published:** 2023-10-17

**Authors:** Diana Alejandra Lares-Gutiérrez, Marisol Galván-Valencia, Irene Jazmín Flores-Baza, Blanca Patricia Lazalde-Ramos

**Affiliations:** Maestría en Ciencia y Tecnología Química, Unidad Académica de Ciencias Químicas, Universidad Autónoma de Zacatecas, Zacatecas 98000, Mexico; 32130452@uaz.edu.mx (D.A.L.-G.); gavm001144@uaz.edu.mx (M.G.-V.); irenefbaza@gmail.com (I.J.F.-B.)

**Keywords:** carbohydrate-free diet, metabolic syndrome, biochemical parameters, morphometric parameters, rat model

## Abstract

Carbohydrate intake restriction positively affects markers related to metabolic syndrome (MS). However, the effects of long-term carbohydrate-free diets (CFD) have yet to be studied. The main objective of this study was to report the effects on biochemical and morphometric parameters in a rat model of MS. Male Wistar rats were initially divided into two groups: the standard diet group (SD, *n* = 20); and the MS group (*n* = 30) fed a high-glucose diet. Ten animals from each group were sacrificed after 20 weeks on their respective diets to verify MS development. The remaining MS animals were divided into two subgroups: one continued with the MS diet (*n* = 10); and the other transitioned to a carbohydrate-free diet (MS + CFD group, *n* = 10) for 20 more weeks. At week 40, parameters, including glucose, insulin, lipid profile, ketone bodies, C-reactive protein (CRP), aspartate aminotransferase (AST), alanine aminotransferase (ALT), urea, creatinine, liver and muscle glycogen, and serum, hepatic, renal, and pancreatic malondialdehyde (MDA) levels were assessed. Transitioning to CFD resulted in decreased caloric intake and body weight, with normalized parameters including MDA, insulin, lipid profile, ALT, liver glycogen, creatinine, and CRP levels. This shift effectively reversed the MS-induced alterations, except for glycemia and uremia, likely influenced by the diet’s high protein content stimulating gluconeogenesis. This research underscores the potential benefits of long-term carbohydrate restriction in mitigating MS-related markers.

## 1. Introduction

Metabolic syndrome (MS) is a global health problem characterized by a cluster of metabolic abnormalities that include abdominal obesity, systemic hypertension, insulin resistance, and atherogenic dyslipidemia, conferring a higher risk of developing cardiovascular disease (CVD) and type 2 diabetes mellitus (T2DM) [1,2]. Studies published in recent years estimate that 20 to 30% of the world’s adult population meets the criteria for MS [3]. According to the Center for Disease Control and Prevention (CDC), the prevalence of MS has increased drastically in the last four decades [4,5].

The pathogenesis of MS involves both genetic and acquired factors [6,7]. The rise in the prevalence of MS is attributed mainly to excessive caloric intake and the lack of physical activity [8,9]. Access to diets with high fat and carbohydrate contents has contributed to the onset and progression of MS [10,11]. However, it has been demonstrated that an increased consumption of carbohydrates is closely associated with the growing prevalence of MS [5].

Dietary modifications, particularly carbohydrate restriction, have proven to be a more effective therapeutic tool in improving MS than diets that restrict the amount of fat [12,13]. The evidence supports that replacing carbohydrates with saturated or unsaturated fats (monounsaturated and polyunsaturated fats) and protein has essential effects on glycemic control, weight loss, and dyslipidemia [14].

Carbohydrate-free diets have been evaluated in models with healthy genetically modified animals susceptible to the development of T2DM, finding an increase in energy expenditures and gluconeogenesis [15], a decrease in body weight and energy intake [13,16], and avoiding the destruction of pancreatic β-cells [17,18]. However, these models lack the complexity of the metabolic abnormalities that occur in MS, showing limited and isolated results.

Additionally, the studies evaluating the effects of carbohydrate-free diets have generally been of short duration, failing to provide a comprehensive understanding of the long-term effects that mirror the chronic dietary exposure seen in individuals predisposed to metabolic syndrome.

Therefore, given the limitations of short-duration studies and the critical need to comprehend the chronic and long-term effects of a carbohydrate-free diet, this investigation aims to address this gap by employing a more extended experimental period, evaluating the effects of chronic exposure to a carbohydrate-free diet on morphometric and biochemical parameters in a rat model of diet-induced MS, evaluating the characteristic alterations of MS.

## 2. Materials and Methods

### 2.1. Animals and Diets

This study used fifty male Wistar rats weaned at 21 days and with an average body weight of 61.6 g. The animals, provided by the bioterium “Claude Bernard” of the Autonomous University of Zacatecas, were housed in polycarbonate cages with a 12-h light/dark cycle, a controlled temperature of 22 °C, and 50% humidity.

The animals were randomly divided into groups according to the type of diet supplied. The standard diet (SD) group (*n* = 20) was fed a balanced commercial diet for rodents (Harlan Teklad Rat Food Diet 8604; Madison, WI, USA), with 14% calories of energy as fat, 32% as protein, and 54% as carbohydrates and fresh water ad libitum. Ten animals were sacrificed after 20 weeks, while the others continued until the 40th week.

The metabolic syndrome (MS) group (*n* = 30) was fed a formulated diet with 16% calories of energy as fat, 33% as protein, 51% as carbohydrates, and 25% as glucose water solution ad libitum. Ten animals were sacrificed at week 20 to confirm the MS state; ten continued the MS diet until week 40, and for the other ten animals, at week 20, the diet was switched to a formulated carbohydrate-free diet providing 52.9% of energy from fat, 47% kcal from protein, 0.1% kcal from carbohydrates, and freshwater ad libitum for 20 weeks (Figure 1).

MS diet was formulated according to the Official Mexican Norm “NOM-062-ZOO-1999”, entitled to technical specifications for producing, caring for, and using laboratory animals [19]. The selection and bromatological composition of the ingredients included in the SM diet were obtained from the Spanish Foundation for the Development of Animal Nutrition FEDNA 2019 [20], and the dose of glucose (25%) administered was based on what was previously described by other authors [21].

The formulated MS diet consisted of the following ingredients (g/kg): fish flour, 250; bentonite, 50; lard, 3.2; rice, 153.3; oatmeal, 340; corn, 100; coconut oil, 3.5; wheat, 100. Once the SM diet was formulated, all the ingredients that provide sugars and assimilable carbohydrates were eliminated to prepare a carbohydrate-free diet (CFD). It consisted of the following(g/kg): fish flour, 450; casein, 200; rice husk, 100; bentonite, 50; lard, 200. Bromatological composition and energy density of the SD, MS, and CDF are shown in Table 1.

Energy density was estimated using the Atwater factor. During the experimental period, everyday food and water or glucose water solution intake was recorded; this information and the energy density were used to estimate the average caloric intake by a group. The body weight of the animals was recorded once a month. The MS state was confirmed by measurement of fasting blood glucose, insulin, lipids profile, and C-reactive protein, as later described.

### 2.2. Samples Collection

After previous anesthesia, the blood of the animals fasted for 12 h was obtained by cardiac puncture in the ventricular cavity. Blood was centrifugated, and serum samples were stored at −20 °C until analysis. The liver, kidney, pancreas, and skeletal muscle were dissected and rinsed in phosphate buffer solution pH 7.

### 2.3. Determination of Biochemical Parameters

Serum concentrations of glucose, total cholesterol (TC), triglycerides (TRIG), high-density lipoprotein cholesterol (HDL-c), low-density lipoprotein cholesterol (LDL-c), alanine aminotransferase (ALT), aspartate aminotransferase (AST), urea, and creatinine were determined by colorimetric enzymatic assays (Spinreact, Girona, Spain). Very low-density lipoprotein cholesterol (VLDL-c) concentration was calculated according to the Equation (1):VLDL-c = TRIG/5,(1)

C-reactive protein (CRP) concentration was determined by a semiquantitative plate agglutination assay (Spinreact, Girona, Spain).

Insulin determination was performed using an ELISA kit, specifically for rats (Spibio, Bertin Pharma, Montigny-le-Bretonneux, France).

β-hydroxybutyrate quantification was determined using reactive strips and a FreeStyle Optium Neo electronic meter (Abbott, Mexico City, Mexico).

### 2.4. Determination of Malondialdehyde

Damage to lipid membranes in blood serum, liver, kidney, and pancreas was assessed by the formation of malondialdehyde (MDA) using the thiobarbituric acid (TBA) reactive method [22]. Tissues were homogenized at 10% KCl (1.15%) and blood serum in H_2_SO_4_ (12 N); the addition of TBA (0.3%) created colorful compounds that were measured photometrically at 534 nm wavelength in a 6715 UV/Vis Jenway spectrophotometer (Cole-Parmer, IL, USA). The MDA level in the samples (expressed as nM/mL in serum and nM/g in tissues) was determined using a calibration curve with 1,1,3,3-tetraethoxypropane (Sigma Chemical, St. Louis, MO, USA).

### 2.5. Determination of Glycogen

The anthrone reagent method determined glycogen levels in the liver and skeletal muscle [23]. A hydrolyzate of liver and skeletal muscle was made in KOH (30%) in a 1:3 (*w*/*v*) ratio; the addition of anthrone solution allowed for the coloration of the sample, which was measured photometrically at 620 nm with a 6715 UV/Vis Jenway spectrophotometer. Glycogen concentration was determined using a glucose (50%) calibration curve (expressed as µg/mL).

### 2.6. Statistical Analysis

Experimental results are shown as mean ± SD or median (minimum–maximum). Statistical analysis was performed with SPSS IBM23 (IBM Corp., Chicago, IL, USA) and GraphPad Prism 5 (GraphPad Software, San Diego, CA, USA) packages. For parametric data, analysis of variance (ANOVA) followed by Tukey’s post hoc analysis was performed. Data not normally distributed were evaluated using the Kruskal–Wallis test and Dunn’s test for post hoc multiple comparisons. Differences with a value of *p* < 0.05 were considered statistically significant, and these were denoted by asterisks as follows: * *p* < 0.05; ** *p* < 0.01; and *** *p* < 0.001.

### 2.7. Ethical Considerations

The handling and care of the animals, as well as their euthanasia, were carried out in accordance with the regulations and the official standards in force, such as the Official Mexican Norm “NOM-062-ZOO-1999,” which shows the specifications and techniques for the production, care, and use of institutional laboratory animals [19]. Animal remains were treated as indicated by the “NOM-087-ECOL-SSA1-2002” for Environmental Protection—Environmental Health—Biologically Infectious Hazardous Waste—Classification and Management Specifications [24].

## 3. Results

### 3.1. Morphometric Parameters and Calorie Intake

Data for food, drink, caloric intake, and body weight are shown in Figure 2. During the first 20 weeks of dietary exposure, the MS group showed a significant decrease in food and drink consumption concerning the SD group. However, the caloric intake in the MS group since week 20 significantly increased compared to the SD and MS + CFD groups; consequently, the body weight increased in the MS group.

The change to CFD in week 20 in the MS rats significantly decreased caloric intake due to lower consumption of solid food and drink compared to the SD and MS groups. The CFD also induced a significant loss in body weight.

In the first 20 weeks of experimentation, the SD and MS groups increased their body weight, the SD group showing the highest body weight. From 20 weeks to the end of the examination period, the MS + CFD and MS groups decreased and increased body weight, respectively, being statistically significant concerning the SD group.

During euthanasia, animals in the MS group exposed to the high-glucose diet for 20 weeks showed a considerable accumulation of visceral fat relative to the SD group, even though the latter group showed higher body weight. At the end of the 40 weeks, the visceral fat in MS animals increased to observed at week 20, while visceral fat in the animals of the MS + CFD group diminished. In both cases, the visceral fat accumulation change was related to the change in body weight.

### 3.2. Biochemical Parameters

#### 3.2.1. Glucose, Insulin, and β-Hydroxybutyrate

The glucose concentration and insulin level comparisons between the experimentation groups are shown in Figure 3. At week 20 of exposure to the diets, the MS group showed a statistically significant increase in glucose concentration (*p* = 0.016) and a decrease in insulin concentration (*p* = 0.008) compared to the SD group. On the other hand, there were no significant differences between the groups in the β-hydroxybutyrate concentrations.

At week 40 of experimentation, the MS diet produced a statistically significant increase in glucose concentration and a decrease in insulin level compared with the SD group (*p* = 0.0001, *p* = 0.023, respectively). When carbohydrates were reduced from the diet (MS + CFD group), a non-significant reduction in glucose level and an increase in insulin concentration were observed compared to the MS group (*p* = 0.0001). The comparison between SD and MS + CFD groups did not show a significant difference in insulin concentration, but there was an increase in glucose concentration (*p* = 0.004). The MS + CFD group presented the highest levels of β-hydroxybutyrate related to the other groups; this difference was only significant with the MS group (*p* = 0.001) (Figure 3).

#### 3.2.2. Lipid Profile

At week 20 of diet exposure, the MS group showed significant differences in all lipid profile parameters (TRIG, TC, HDL-c, LDL-c, and VLDL-c) compared to the SD group (*p* = 0.009, *p* = 0.009, *p* = 0.027, *p* = 0.009, and *p* = 0.009, respectively), as shown in Figure 4.

At week 40 of experimentation, the MS and the SD groups showed a similar behavior than in the previous evaluation, showing significant differences between both in TRIG, TC, HDL-c, LDL-c, and VLDL-c (*p* = 0.007, *p* = 0.0001, *p* = 0.0001, *p* = 0.0001, and *p* = 0.007, respectively), as shown in Figure 4. The MS + CFD group presented a normalization of the lipid profile parameters, showing no significant differences with respect to the SD group.

#### 3.2.3. C-Reactive Protein

Regarding the CRP concentrations, at week 20 of diet exposure, the MS group showed a significant increase related to the SD group (*p* = 0.005). At week 40 of experimentation, the MS and MS + CFD groups showed a significant increase compared to the SD group (*p* = 0.0001; *p* = 0.024, respectively). However, the MS + CFD group showed a non-significant 50% decrease relative to the MS group (Figure 5).

#### 3.2.4. Serum Urea and Creatinine

The MS + CFD group increased the urea concentration by 42.6% in relation to the SD group, this being statistically significant (*p* = 0.004). The MS group increased by 36.5% in relation to the SD group, showing no significant differences in relation to the MS + CFD group. The MS group showed an increase in creatinine level by 66.3% compared to the SD group and by 112.1% in relation to the MS + CFD group, this difference being significant only in relation to the MS + CFD group (*p* = 0.0001) (Figure 6).

#### 3.2.5. Hepatic Enzymes and Glycogen

The MS + CFD group showed a significant decrease in AST concentration by 47.05% in relation to the SD group and by 40.7% in relation to the MS group, these differences being statistically significant (Table 2). Regarding ALT concentration, the MS group showed the lowest levels compared to the SD and MS + CFD groups, these differences being statistically significant (Table 2).

The MS group presented the highest concentrations of liver glycogen compared to the SD and the MS + CFD groups, the difference with respect to the SD group being statistically significant, while the MS + CFD group showed a non-significant decrease of 51% compared to the MS group. At the skeletal muscle level, no significant differences were observed between the three groups of study; however, the MS group showed the highest concentrations of muscle glycogen in relation to the SD and MS + CFD groups (Table 2), the SD and MS + CFD groups showed concentrations medians very similar.

#### 3.2.6. Malondialdehyde

Figure 7A shows the behavior of the serum MDA concentration in the MS group, presenting a statistically significant increase in relation to the SD and MS + CFD groups (*p* = 0.013 and *p* = 0.005, respectively) with increases of 40.9 and 39.8%, respectively.

At the hepatic level, the MS group presented a significant increase by 100.4% in relation to the SD group (*p* = 0.041), and the MS + CFD group showed a non-significant 30.3% increase in relation to the SD group and a non-significant 35% decrease relative to the MS group (Figure 7B).

At the renal level, no significant differences in MDA concentrations were observed between the three study groups; however, the MS group showed a 26.6% increase in relation to the SD group; the MS + CFD group presented a 41.8% increase relative to the SD group (Figure 7C).

At the pancreatic level, the MS and MS + CFD groups showed a non-significant decrease in MDA concentrations by 35.5% and 20.9%, respectively, in relation to the SD group (Figure 7D).

## 4. Discussion

MS is a group of several metabolic disorders that raise the risk of developing heart disease and diabetes, among other illnesses. The etiology of MS is multifactorial, with environmental and genetic components, whence the pharmacological treatments and lifestyle changes in the MS have become a challenge.

The association between the prevalence and incidence of MS worldwide and increased unhealthy food consumption has stimulated the study of new dietary alternatives for MS treatment as well as the generation of diet-induced MS models.

Diet-induced MS models are used because of their simplicity and low cost. High-carbohydrate (fructose and sucrose) and high-fat diets are the most employed models. Previous investigations have demonstrated more pronounced alterations in rats fed high-fat/high-glucose diets compared to those fed fructose-based diets. Furthermore, it is suggested that adopting a healthy eating pattern significantly reduces the risk of developing MS. Nevertheless, adequate nutritional intervention is still controversial. The reduction in the intake of carbohydrates in the diet is one of the main recommendations to reduce body weight, glycemia, and dyslipidemia [25]. However, diets that entirely restrict carbohydrate intake have yet to be evaluated in animal models of MS, where the parameters that make up this condition are present jointly.

The diagnostic criteria of MS have not been established for animal models; however, according to a review, three of the following five metabolic parameters can be used in rodent models: increased fasting glucose; obesity; increased triglyceride concentration; decreased HDL-c concentration; and increased systolic blood pressure [26].

Sexual dimorphism is essential in animal research, primarily due to hormonal variations between males and females. Studies have shown that the repercussions of diet-induced metabolic alterations differ between sexes [27]. Gonadal hormones and genetic factors are intertwined with fat distribution in animal models. Female animals typically exhibit a higher proportion of subcutaneous fat and a lower amount of visceral fat than males [28]. Research demonstrates that male rodents display greater susceptibility to developing MS when induced by high-calorie diets, manifesting more pronounced alteration in glucose tolerance, hyperinsulinemia, hyperleptinemia, and weight gain compared to females [29].

Previous reports in which diets with high sugar in drinking water induced MS have demonstrated changes in glucose homeostasis and insulin, oxidative imbalance, low-grade inflammation, and plasma dyslipidemia [30,31].

In our work, the animals that consumed the formulated solid diet and a 25% glucose solution significantly increased fasting glucose, triglycerides, HDL-c, and CRP concentrations and decreased insulin concentration since the first 20 weeks in relation to the SD group. After 20 weeks, the diet increased food and glucose solution consumption. Consequently, calorie intake and body weight increased. Excess calorie intake is essential in developing metabolic complications related to MS [26]. This finding suggests alterations in hunger and satiety mechanisms that regulate food intake after a long-term administration of the MS diet. At the end of the 40th week, the biochemical alterations remained in the MS group.

An increase in visceral fat deposition in diet-induced MS animal models is a determining factor in the establishment of obesity in rodents as it is related to the development and progression of MS [32]. At week 20, the animals in the MS group showed a considerable increase in visceral fat deposits compared to the SD group, coinciding with publications in which exposure to high-carbohydrate diets led to visceral fat accumulation [33,34].

The MS + CFD group significantly reduced water and food consumption at 40 weeks. The decrease in caloric intake observed after exposure to CFD with high protein content is consistent with previous reports in animals. These have been related to a state of satiety and increased energy expenditure produced by high-protein diets [13,16,35]. In this sense, the decrease in caloric intake can be related to the decrease in body weight, possibly through an increase in energy expenditure [36].

Previous studies directly correlate the carbohydrate content in the diet with the increase in body weight and adiposity since carbohydrates are the main secretagogue of insulin, stimulating fat storage [16].

According to certain reviews concerning animal models of diet-induced MS, significant increases in the animal’s body weight are only occasionally observed in comparison to control groups, even when they exhibit significant metabolic alterations [37,38]. Most of the studies reported exposure to diets between 6 and 20 weeks. In this study, significant increases in body weight are observable in animals with MS starting from week 20 of experimentation, in contrast to the SD group. This outcome confirms the impact of chronic exposure to a high-glucose diet on body weight gain in animal models.

Insulin resistance serves as a key connecting factor between excessive carbohydrate consumption and the development of MS [39]. Insulin sensitivity assessment was conducted by measuring fasting glucose and insulin levels at the 40th week of dietary exposure. In the MS group, the high-glucose diet led to disruptions in carbohydrate metabolism, particularly the emergence of both hepatic and systemic insulin resistance, evident as hyperglycemia due to the failure to suppress glucose production in the liver. Additionally, this same group exhibited a significant decrease in insulin concentration compared to the SD group. Although hyperinsulinemia is considered a significant predictor of MS development, the inability to secrete insulin is strong evidence of pancreatic β-cell deterioration to external stimuli, such as high glucose concentrations for prolonged periods [40].

The CFD (containing high fats and proteins) caused a non-significant decrease in glucose concentration compared to the MS group, but it did not reach the level of normalization observed in the SD group. The CFD induced notable changes in carbohydrate metabolism due to the strict requirement of glucose by the central nervous system and other tissues, leading to the activation of hepatic gluconeogenesis. These findings align with results previously reported by other authors [41,42]. The high energy demands of the gluconeogenesis pathway require a constant flow of ATP, which is facilitated by increased fatty acid oxidation. This phenomenon explains the weight loss observed in response to this pathway’s upregulation [43]. Similarly, the insulin concentration in the MS + CFD group showed a normalization comparable to the SD group. There are reports about the role of carbohydrate-restricted diets in reducing oxidative stress and pro-inflammatory conditions, as well as their positive impact on maintaining pancreatic β-cell integrity in the early stages of T2DM in animal models [44].

Ketone bodies can serve as substitutes for glucose in conditions of fuel and food deficiency. β-hydroxybutyrate is the ketone body found in the highest concentration in blood. It has a fundamental role in the temporary inhibition of lipolysis, inflammation, and satiety and in addressing conditions like atherosclerosis and oxidative stress [45]. The reduction in dietary carbohydrates promoted the activation of the ketogenesis pathway, utilizing free fatty acids as an energy source. It has been reported that a high concentration of β-hydroxybutyrate promotes satiety when a high-protein diet is implemented [35]. Additionally, β-hydroxybutyrate treatment in mice exposed to a high-fat diet has demonstrated the ability to maintain metabolic health by influencing pathways related to glucose homeostasis, mitochondrial function, and obesity [46].

While existing research has underscored the beneficial effects of ketogenic diets (characterized by high-fat content with a regular protein content) primarily attributable to the role of β-hydroxybutyrate in metabolic pathways [47,48], our findings revealed a notable deviation from the expected ketogenic state, as evidenced by the absence of a significant increase in ketone bodies compared to the SD group. Supporting the modification of other pathways, such as the reduction in the inflammatory state and obesity, as well as oxidative stress as the main pathways to improve the metabolic alterations of MS.

Dyslipidemia is an essential alteration of MS and the leading cause of the development of cardiovascular disease. Abnormalities in the lipid profile have been linked to visceral obesity and insulin resistance [49]. According to previous authors, lipid markers show improvement when saturated, monounsaturated, or polyunsaturated fats replace carbohydrates in the diet [50]. The role of fats in the diet has an erroneous assumption regarding weight gain, primarily due to the differing energy density of the macronutrients: fats provide 9 kcal/g compared to 4 kcal/g from both protein and carbohydrates. However, it is common to overlook the role of macronutrients in regulating hunger, satiety, and the pathways that control energy generation, fat storage, and fatty acid metabolism [51].

The MS group exposed to the high-glucose diet showed dyslipidemia characterized by significant alterations in triglycerides and HDL-cholesterol concentrations. In contrast, the MS + CFD group presented a normalization of lipid profile parameters despite an increased dietary fat content within the CFD.

The observed results suggest that the almost elimination of carbohydrates in the diet promotes the oxidation of lipids at the expense of their deposits; thus, the macronutrients provided by the meals are entirely oxidized after ingestion, serving as the primary source for energy generation (including gluconeogenesis and ketogenesis), reflected in a reduction in body weight as was observed in the MS + CFD group about the MS group fed with a high-glucose diet. This hypothesis is supported by previous studies demonstrating decreased de novo liver lipogenesis in animals fed a high-protein, carbohydrate-free diet [13,52]. Consequently, it is plausible that the reduction in liver lipogenesis and the resulting decreased transport of triglycerides to extrahepatic tissues over an extended period contributes to a lower body fat content.

The characteristic visceral obesity observed in MS causes a condition of hypertrophy and hypoxia within adipocytes, leading to necrosis and infiltration of macrophages into adipose tissue. This, in turn, results in an overproduction of pro-inflammatory cytokines, such as tumor necrosis factor-alpha (TNF-α), interleukin-6 (IL-6), and C-reactive protein (CRP), thereby triggering systemic inflammation. CRP, an acute-phase reactant, is produced by hepatocytes in response to IL-6 stimulation [53]. In the MS group exposed to the high-glucose diet, CRP concentrations increased compared to the SD group. According to several reports, elevated CRP levels have been directly correlated with conditions like obesity, dyslipidemia, hyperglycemia, and insulin resistance [54,55].

After exposure to CFD, the animals improved their systemic inflammatory state. A direct relationship between dietary carbohydrate restriction and improved inflammation has been described, mediated by adenosine levels, ketone bodies, caloric restriction, and gut microbiota [56]. Additionally, the MS + CFD group showed a notable reduction in visceral fat deposits concerning the MS group, although evidence was not included in this report. Moreover, the CFD reduced adipocyte size, unlike the MS group, where hypertrophic adipocytes were observed. The normalization of adipocyte size in the MS + CFD group is associated with improving the inflammatory state. This effect is likely achieved by suppressing the production of pro-inflammatory cytokines implicated in developing conditions such as insulin resistance.

MS, particularly obesity, insulin resistance, and inflammation, are associated with alterations in liver function. Several reports indicate that liver function, particularly the levels of total bilirubin, ALT, and AST, are closely related to the incidence of MS. These parameters have also been identified as predictive factors of the development of hepatic steatosis and non-alcoholic fatty liver disease (NAFLD) [57].

Previous clinical studies incorporating a partially restricted carbohydrate diet reported no changes in the concentrations of these enzymes in obese patients with T2DM [58,59] and patients with NAFLD [60]. However, in our study, the concentration of AST and ALT decreased in the MS + CFD group; this may be because carbohydrate restriction is related to states of ketosis observed in this group. Ketosis is associated with increased blood lactate, which causes a rapid consumption of NADH, related to a decrease in the enzymatic activity of AST and ALT [61].

Glycogen is an energy reservoir stored in the liver and skeletal muscle. Liver glycogen is the main store that maintains blood glucose homeostasis. Alterations in glucose homeostasis within the context of MS significantly affect the levels of this polymer [62]. It has been demonstrated that during the initial hours of exposure to a carbohydrate-free and high-protein diet, glycogenolysis is the primary pathway for endogenous glucose production. However, once glycogen levels decrease, the gluconeogenesis pathway becomes the predominant route [15]. In the present study, the elimination of carbohydrates from the diet of the MS + CFD group induced a decrease in glucose stored as glycogen in the liver and skeletal muscle when compared to the MS group during the activation of glycogenolysis for energy requirement.

MS is associated with an increased risk of developing kidney abnormalities. According to some studies, hypertension, hyperglycemia, dyslipidemia, and obesity are pivotal in promoting kidney damage [63]. Although there were no significant differences between the SD and MS groups, the latter group showed an increased tendency in urea and creatinine concentrations, demonstrating the role of a high-glucose diet in generating renal alterations. Possible mechanisms by which MS affects renal physiology include impaired renal hemodynamics, insulin resistance, hyperlipidemia, activation of the renin–angiotensin–aldosterone system, inflammation, and oxidative stress [64].

On the other hand, the high protein content in CFD might have fundamentally impacted the metabolic processes regulated by the kidneys and renal function [65]. Increased urea excretion in the MS + CFD group about the SD group was observed; however, creatinine levels were not affected by exposure to CFD, thus indicating that the activation of the gluconeogenesis pathway for energy generation occurred from protein amino acids from the diet and not from the skeletal muscle. These results agree with previous studies, who report no changes in the creatinine concentration in animals subjected to a high-protein and low-carbohydrate diet [65]. There is evidence about the relationship between diets low in carbohydrates and high in protein and fat and the increase in the glomerular filtration rate, possibly reflected in kidney damage [66,67]. Nevertheless, previous authors suggest the increase in serum urea as a standard adaptive mechanism that occurs in response to an increase in protein metabolism and does not cause adverse effects on renal morphology [68].

Oxidative stress, whose etiology is mainly mitochondrial dysfunction, is related to the development of MS, causing inflammation, thrombosis, and atherosclerosis [69]. MDA concentration was evaluated as a marker of oxidative damage (lipoperoxidation) in blood serum, liver, kidney, and pancreas. In the MS group, the high glucose content in the diet probably led to an increase in oxidative activity, causing an overproduction of reactive oxygen species (ROS) in the mitochondria, generating damage to macromolecules [69], reflecting in increases in MDA concentrations at serum and liver levels. Diverse studies have reported increased serum and liver MDA concentrations after exposure to a high-carbohydrate diet; the authors relate this increase with an inflammatory state related to high CRP levels [33,70].

In contrast, the MS + CFD group showed a decrease in serum MDA concentration, indicating a decrease in oxidative activity and ROS production in the mitochondria [71]. Similarly, decreased lipid peroxidation is related to increased levels of β-hydroxybutyrate since it inhibits pathways related to the production of ROS and the stimulation of antioxidant enzyme activity [46,72]. The results are consistent with those previously observed, where carbohydrate-restricted diets are incorporated [72].

This study has several limitations. First, only the effect of the carbohydrate-free diet on oxidative damage at the lipid level was evaluated, but the oxidative damage on protein and DNA was not. Second, evaluating the effect of the carbohydrate-free diet on animals that previously received the SD diet is necessary. Third, no behavioral changes due to the absence of carbohydrates in the diet were evaluated in this study. Fourth, incorporating both sexes in the experimental model can enhance the translational value and broaden our understanding of MS-related mechanisms and potential interventions. Finally, this study evaluated the biochemical and somatometric effects of the chronic administration of a carbohydrate-free diet; however, we need to find out the short-term effects of this diet.

## 5. Conclusions

In conclusion, our study demonstrates that chronic administration of a carbohydrate-free diet in a rat model of diet-induced metabolic syndrome resulted in significant improvements in biochemical and morphometric parameters. The CFD led to decreased caloric intake and body weight, normalization of serum and hepatic lipid peroxidation, insulin, lipid profile, ALT enzyme activity, liver glycogen, creatinine, and decreased inflammatory state. However, glycemia and uremia did not significantly improve, probably due to the high protein content in the diet.

Further research is needed to understand the underlying mechanisms responsible for the observed improvements.

## Figures and Tables

**Figure 1 metabolites-13-01085-f001:**
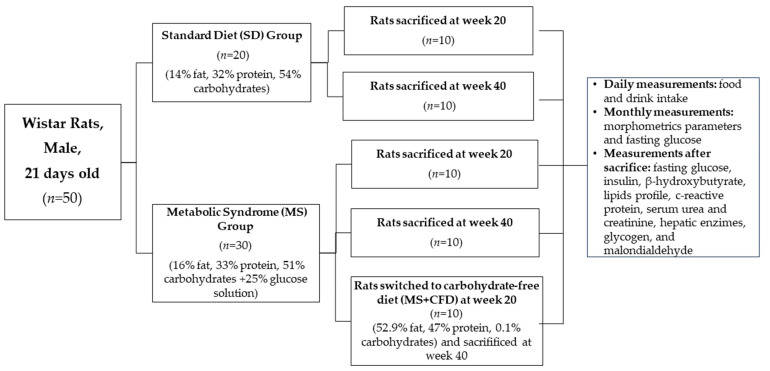
Experimental protocol diagram.

**Figure 2 metabolites-13-01085-f002:**
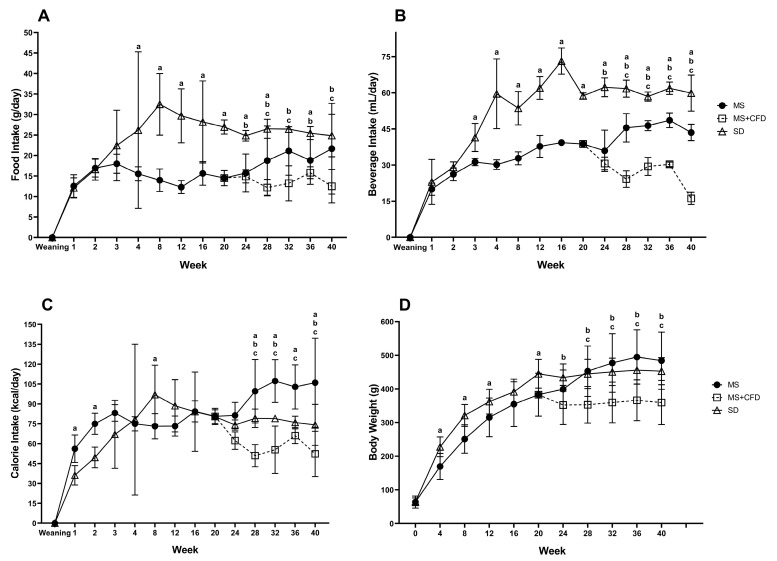
(**A**) Food and (**B**) beverage intake, (**C**) total caloric intake, and (**D**) body weight in the groups studied throughout the 40 weeks of experimentation. Data are presented as means ± SD. The comparison between groups was performed using an analysis of variance (ANOVA), followed by Tukey’s post hoc adjustment test. It was considered statistically significant when *p* < 0.05. The statistical differences between groups are presented as a: SD vs. MS, b: SD vs. MS + CFD, and c: MS vs. MS + CFD. Abbreviations: SD, standard diet; MS, metabolic syndrome; MS + CFD, metabolic syndrome and subsequent carbohydrate-free diet.

**Figure 3 metabolites-13-01085-f003:**
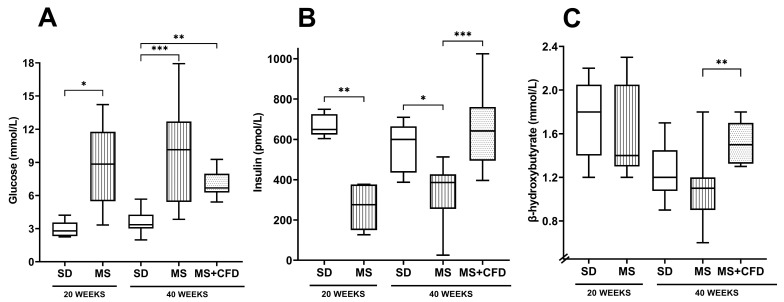
(**A**) Glucose, (**B**) insulin, and (**C**) β-hydroxybutyrate concentrations at the 40th week of experimentation in the study groups. Data are presented as median (minimum–maximum). Comparison analysis between groups was performed using the Kruskal–Wallis test and Dunn’s adjustment post hoc test. It was considered statistically significant when *p* < 0.05, and these were denoted by the following asterisks: * *p* < 0.05; ** *p* < 0.01; and *** *p* < 0.001. Abbreviations: SD, standard diet; MS, metabolic syndrome; MS + CFD, metabolic syndrome and subsequent carbohydrate-free diet.

**Figure 4 metabolites-13-01085-f004:**
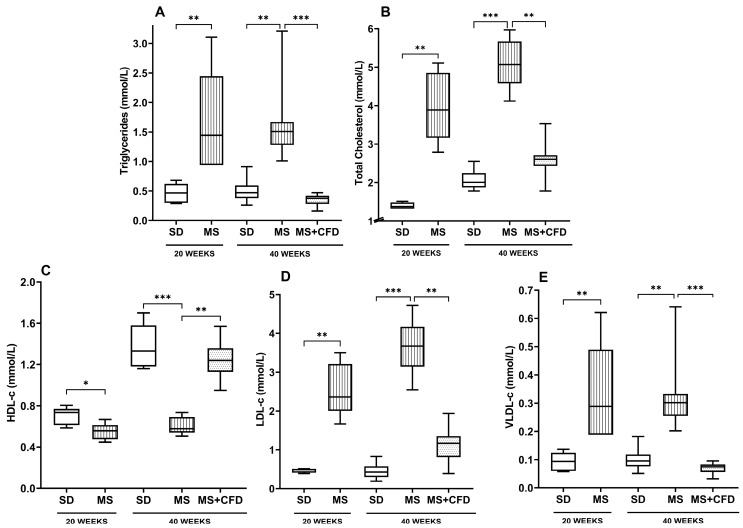
(**A**) Triglycerides, (**B**) total cholesterol, (**C**) HDL-c, (**D**) LDL-c, and (**E**) VLDL-c concentrations at the 40th week of experimentation in the study groups. Data are presented as median (minimum–maximum). Comparison analysis between groups was performed using the Kruskal–Wallis test and Dunn’s adjustment post hoc test. It was considered statistically significant when *p* < 0.05, and these were denoted by the following asterisks: * *p* < 0.05; ** *p* < 0.01; and *** *p* < 0.001. Abbreviations: SD, standard diet; MS, metabolic syndrome; MS + CFD, metabolic syndrome and subsequent carbohydrate-free diet.

**Figure 5 metabolites-13-01085-f005:**
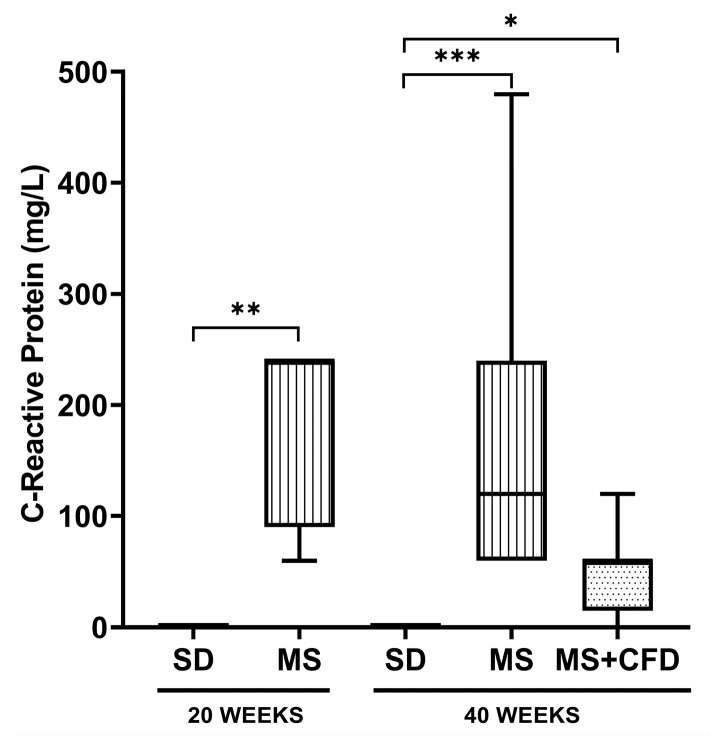
C-reactive protein concentrations in the study groups at the 40th week of experimentation. Data are presented as median (minimum–maximum). Comparison analysis between groups was performed using the Kruskal–Wallis test and Dunn’s adjustment post hoc test. It was considered statistically significant when *p* < 0.05, and these were denoted by the following asterisks: * *p* < 0.05; ** *p* < 0.01; and *** *p* < 0.001. Abbreviations: SD, standard diet; MS, metabolic syndrome; MS + CFD, metabolic syndrome and subsequent carbohydrate-free diet.

**Figure 6 metabolites-13-01085-f006:**
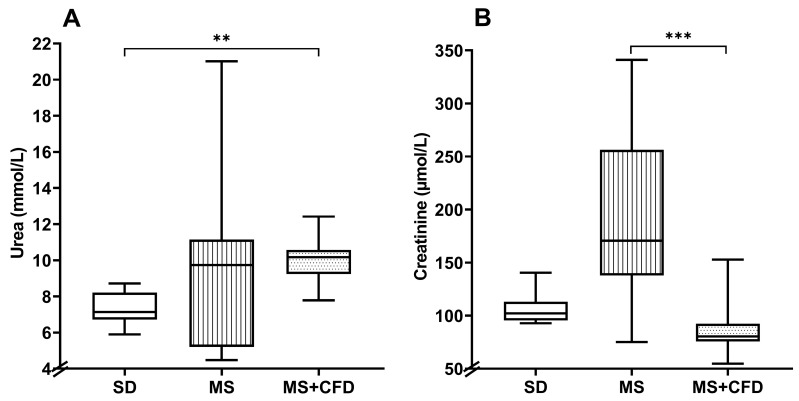
(**A**) Serum urea and (**B**) serum creatinine concentrations at the 40th week of experimentation in the study groups. Data are presented as median (minimum–maximum). Comparison analysis between groups was performed using the Kruskal–Wallis test and Dunn’s adjustment post hoc test. It was considered statistically significant when *p* < 0.05, and these were denoted by the following asterisks: ** *p* < 0.01; and *** *p* < 0.001. Abbreviations: SD, standard diet; MS, metabolic syndrome; MS + CFD, metabolic syndrome and subsequent carbohydrate-free diet.

**Figure 7 metabolites-13-01085-f007:**
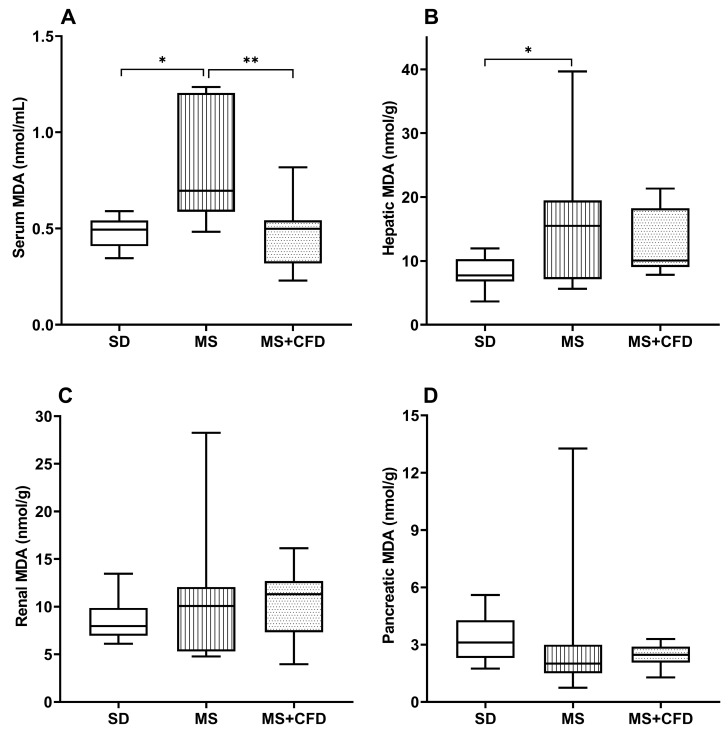
MDA concentrations in (**A**) blood serum, (**B**) liver, (**C**) kidney, and (**D**) pancreas in the study groups at the 40th week of experimentation. Data are presented as median (minimum–maximum). Comparison analysis between groups was performed using the Kruskal–Wallis test and Dunn’s adjustment post hoc test. It was considered statistically significant when *p* < 0.05, and these were denoted by the following asterisks: * *p* < 0.05 and ** *p* < 0.01. Abbreviations: SD, standard diet; MS, metabolic syndrome; MS + CFD, metabolic syndrome and subsequent carbohydrate-free diet; MDA, malondialdehyde.

**Table 1 metabolites-13-01085-t001:** Bromatological composition and energy value of the diets used in this study.

Nutrients	SD Diet			MS Diet			CFD Diet		
	Content (g%)	Kcal (%)	Energy Density (Kcal/g)	Content (g%)	Kcal (%)	Energy Density (Kcal/g)	Content (g%)	Kcal (%)	Energy Density (Kcal/g)
Crude protein	24.30	97.20 (32.3)	0.972	23.87	95.48 (33.1)	0.955	49.16	196.64 (47.05)	1.966
Lipids	4.70 ^1^	42.90 (14.3)	0.429	5.34 ^2^	48.06 (16.0)	0.481	24.59 ^2^	221.31 (52.95)	2.213
Carbohydrates	40.20	160.80 (54.4)	1.608	36.80	147.20 (50.9)	1.472	0.11	0.44 (0.11)	0.004
Crude fiber	4.00	-	-	13.05	-	-	7.57	-	-
Insoluble fiber									
Ashes	7.40	-	-	4.50	-	-	7.49	-	-
Moisture	-	-	-	9.39	-	-	5.88	-	-
Glucose solution	-	-	-	25.00	100	1.0 kcal/mL	-	-	-
Energy density of each diet			3.009 kcal/g			3.908 kcal/g mL			4.183 kcal/g

^1^ Of which 2.1% are polyunsaturated, 1.1% are monounsaturated, and 0.9% are saturated; 50 mg of cholesterol/kg of food. ^2^ Of the fat content, 0.31% is lard, which provides 0.3 mg of cholesterol/100 g of food.

**Table 2 metabolites-13-01085-t002:** Hepatic enzymes and liver and muscle glycogen concentrations.

Biochemical Parameter	SD Group	MS Group	MS + CFD Group	*p*-Value
AST (U/L)	84.29 (72.92–214.08)	75.25 (54.83–155.75)	44.63 (11.66–88.08)	0.0008 ^2^ 0.013 ^3^
ALT (U/L)	49.88 (39.08–99.17)	16.33 (13.42–24.50)	31.79 (7.00–66.50)	0.0001 ^1^ 0.021 ^3^
Liver glycogen (µmol/L)	1.82 (1.41–2.85)	6.47 (2.69–22.33)	3.13 (0.14–5.23)	0.0001 ^1^
Muscle glycogen (µmol/L)	2.64 (1.51–6.0)	4.23 (1.84–5.75)	2.85 (0.22–5.49)	ns

Data are presented as median (minimum–maximum). A comparison analysis between groups was performed using the Kruskal–Wallis test and Dunn’s adjustment post hoc test. It was considered statistically significant when *p* < 0.05. The comparisons were as follows: ^1^: ND vs. MS; ^2^: ND vs. MS + CFD; ^3^: MS vs. MS + CFD. Abbreviations: SD, standard diet; MS, metabolic syndrome; MS + CFD, metabolic syndrome, subsequent carbohydrate-free diet; ALT, alanine aminotransferase; AST, aspartate aminotransferase; ns, non-significant.

## Data Availability

Not applicable.

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
