# Peer review of "Benefits of Chronic Administration of a Carbohydrate-Free Diet on Biochemical and Morphometric Parameters in a Rat Model of Diet-Induced Metabolic Syndrome"

_metabolites, 2023, doi:10.3390/metabo13101085_

Round 1

Reviewer 1 Report

Carbohydrate restriction may have beneficial effects on health. In their manuscript, using a rat model, Lares-Gutierrez et al submitted animals to either a standard chow diet, a “metabolic syndrome” diet and a carbohydrate-free diet, of the same composition of the MS diet, but with removal of carbohydrates.

Diet duration was 40 weeks, and after 20 weeks, part of the animals on the MS diet were switched to the CFD diet.

The results presented in the manuscript support a beneficial role of the CFD diet on multiple metabolic parameters,

One drawback of the study is that the authors do not attempt to identify possible metabolic pathway modulated by the CFD diet. However, the paper, as presented is well performed, informative and will contribute to move forward the field.

I wish to propose some changes that may improve the readability of the manuscript:

1)    For ease of understanding, Description of diets and experimental protocol in 2.1 should be integrated by a schematic diagram.

2)    The experimental change of a diet, as performed here, is a useful and seldomly explored model, that has the advantage of evaluating physiological changes occurring after the installment of a metabolic syndrome phenotype. The authors should note, in their discussion, that some literature using such dietary switch already appeared in mouse models (e.g; PMID 17299079, or, more recently, PMID 35995402).

3)    HGD is mentioned only in the abstract. This should be replaced with MS diet, used uniformly in the manuscript, for consistence.

4)    Often in the manuscript (for example line 216), authors report p=0.000 . This has no real statistical sense because it indicates “certainty”, which cannot apply to experimental research. Please use p<0.0001. Also, figures present the statistical differences levels with asterisks (*, ** etc). I guess this means *< 0.05; **<0.01 etc. Please state this in the 2.6 section and in the figure legends.

Minor editing.

Reviewer 2 Report

The paper entitled "Benefits of chronic administration of a carbohydrate-free diet on biochemical and morphometric parameters in a rat model of diet-induced metabolic syndrome” is an important study that proposes carbohydrate-free diets  to improve metabolic syndrome.

The study is interesting, but there are only some points that require improvements

Methods: 

The animals used for the experiments are only males. This is a crucial bias because males and females could differently respond to the same stimulus due to the different metabolic composition. The results cannot be generalized. Please modify accordingly and add a sentence in the limitations paragraph

Experimental design: It would be useful for the reader to have a scheme or a figure of the entire experimental design

Reviewer 3 Report

This study investigates the impact of long-term carbohydrate-free diets (CFD) on a metabolic syndrome (MS) rat model. Different from former studies, this research robustly designs a time span of 20 and 40 weeks, providing substantial data for evaluating long-term CFD's influence on MS. The methods are precise and standardized, ensuring accurate measurements of morphological and biochemical parameters. The results section offers a thorough analysis, providing reasoned interpretations for parameter trends. It emphasizes long-term CFD's potential benefits in blood glucose control, weight loss, and lipid regulation, supporting further research and clinical applications. The writing is also scientifically accurate and easy for readers to understand. Thus, I recommend this research to be published in Metabolites. Here is a minor suggestion for the authors to consider: This experiment established three control groups (SD, MS, MS+CFD), but the abstract did not effectively introduce these three control groups. This has resulted in an unclear meaning of the abstract, which may lead to confusion for the readers.

Minor revision needed.
